# Secondary findings in a large Pakistani cohort tested with whole genome sequencing

Aliaksandr Skrahin[1] , Huma Arshad Cheema[2], Maqbool Hussain[3], Nuzhat Noureen Rana[4], Khalil Ur Rehman[5] , Raman Kumar[6], Gabriela Oprea[1], Najim Ameziane[1], Arndt Rolfs[1,7], Volha Skrahina[1]

**Studies on genomic secondary findings (SFs) are diverse in participants' characteristics, sequencing methods, and versions of the ACMG SF list. Based on whole genome sequencing and the version 3.1 of the ACMG SF list, we studied SFs in 863 individuals from five different regions in Pakistan. We identified 24 ACMG SFs in 23 (2.7%) of 863 individuals: 18 of 24 were related to cardiovascular disease and four to cancer syndromes. In addition to ACMG SFs, we identified 16 (1.9%) participants with pathogenic and likely pathogenic variants in genes that were not related to the participants' clinical conditions but with clear medical actionability (non-ACMG SFs): 4 of 16 were related to eye diseases, two to metabolic disorders, and two to urinary system disorders. By testing a large Pakistani cohort with whole genome sequencing, we concluded that in countries such as Pakistan, the ACMG SF list could be expanded, and our non-ACMG SF list is one example.**

## Introduction

The comprehensive utility of exome and genome sequencing in clinical practice is accompanied by medical and ethical issues related to reporting on the findings. The diagnosis of a genetic disease requires a comprehensive clinical assessment of the subject, addressing physical features, clinical and laboratory tests, and medical family history. The causative genetic variants that fully or partially explain an individual's clinical condition that prompted the genetic test is referred to as primary findings (PFs). A pathogenic or likely pathogenic variant in a gene, which is not related to the individual's clinical condition that may nonetheless be of medical value to the patient, is referred to as a secondary finding. In 2013, the American College of Medical Genetics and Genomics (ACMG) published recommendations for reporting incidental findings in clinical sequencing for a list of 56 genes (v1.0) associated with Mendelian disorders that are medically actionable (Green et al,

2013). These gene-disease pairs were selected based on clinical evidence for disease severity, high penetrance (i.e., a high probability that the pathogenic variant will express the associated condition, despite symptoms being absent at the time of the test request), and medical actionability (i.e., effective preventive measures and/or treatments are available). In addition, evidence-based guidelines were recommended by ACMG and the Association for Molecular Pathology in 2015 to standardize the clinical interpretation of sequence variants (Richards et al, 2015). Furthermore, the term "incidental findings" was replaced by the more meaningful term "secondary findings" (SFs). The gene list was increased to 59 (v2.0) (Kalia et al, 2017), 73 genes (v3.0) (Miller et al, 2021), and currently includes 78 genes (v3.1) (Miller et al, 2022). A new framework to update the ACMG SF list annually was also outlined.

Studies in the SFs area are diverse in design; number of participants; participants' characteristics such as consanguinity, ethnicity, and symptoms; the use of various sequencing methods (whole genome [WGS] or exome sequencing [WES]); and ACMG SF gene list versions (v1.0 SF, v2.0 SF, and v3.0 SF).

One of the main problems in applying the ACMG SF guidelines is that, to date, expert opinion on the inclusion of these gene-disease pairs in the ACMG SF list is not based on testing of broader patient populations.

Countries and regions that have specific characteristics in the epidemiology of genetic diseases may have criteria for SFs that differ from those recommended by the ACMG. In clinical practice and research, based on frequency of the genetic disorders in the country, the list of SFs can be expanded using the opinions of experts familiar with the regional situation and other relevant resources, e.g., ClinGen and eMERGE. ClinGen (https://www.clinicalgenome.org/) is a National Institutes of Health–funded platform dedicated to building an authoritative central resource that defines the clinical relevance of genes and variants for use in precision medicine, and research resulted in evidence-based reports with semi-quantitative metric scores for 252 clinically actionable genes. eMERGE (https://www.genome.gov/Funded-Programs-Projects/Electronic-Medical-Records-and-Genomics-Network-eMERGE) is a network organized by the National Human Genome Research Institute that combines DNA biorepositories with electronic medical record systems for large scale, high-throughput

---

[1]Arcensus GmbH, Rostock, Germany   [2]University of Child Health Sciences, the Children's Hospital, Lahore, Pakistan   [3]Pakistan Institute of Medical Sciences, Islamabad, Pakistan   [4]The Children's Hospital and the Institute of Child Health, Multan, Pakistan   [5]Town Women and Children Hospital, Peshawar, Pakistan   [6]Liaquat National Hospital, Karachi, Pakistan   [7]University of Rostock, Medical Faculty, Rostock, Germany

Correspondence: aliaksandr.skrahin@arcensus-diagnostics.com

**Table 1. Pakistani cohort characterization.**

| Analysed individuals (n = 863) | Total counts | % |
|---|---|---|
| Index patients | 700 | 81.11% |
| Relatives | 163 | 18.89% |
| **Sex (n = 863)** | Total counts | % |
| Males | 532 | 61.65% |
| Females | 331 | 38.35% |
| **Clinical status (n = 863)** | Total counts | % |
| Clinically affected | 741 | 85.86% |
| Healthy/asymptomatic | 122 | 14.14% |
| **Consanguinity status (n = 700 index patients/families)** | Total counts | % |
| Consanguineous parents | 559 | 79.86% |
| Non-consanguineous parents | 122 | 17.43% |
| Not disclosed | 19 | 2.71% |
| **Age at genetic testing (n = 863), expressed as years** | Mean (median) | SD (range) |
| Clinically affected/symptomatic (n = 741) | 4.93 (3) | 6.17 (0–65) |
| Healthy/asymptomatic (n = 122) | 28.34 (29.5) | 10.52 (0–55) |

genetic research in support of implementing genomic medicine. On the eMERGEseq platform, 109 genes that allow for clinically actionable pathogenic variants to be returned were identified and validated.

Pakistan is a developing country with an estimated population of over 220 million and a high burden of non-communicable diseases. The rate of consanguineous marriages in Pakistan is estimated to be 46–98%, depending on the region. This is associated with high rates of genetically inherited diseases and infant mortality (Riaz et al, 2019). Unfortunately, in developing countries, access to genetic testing is limited (Thong et al, 2018).

Cheema et al (Cheema et al, 2020) reported the results of genetic testing of more than 1,000 individuals from Lahore (Pakistan). The study, which aimed to establish the genetic diagnosis in patients with suspected genetic diseases, demonstrated high diagnostic yield (61%) and clinical impact (change in management in 52% patients). However, the study was focussed on primary genetic findings and limited to the population of the Lahore region. Furthermore, the most frequently performed test (78% of index cases) was WES.

Taking advantage of WGS over WES and the latest version (v3.1) of the ACMG SF list over previous versions, we explored the prevalence of SFs in individuals clinically suspected of genetic diseases from five geographical regions in Pakistan.

In addition to the ACMG SF gene list, we have identified pathogenic and likely pathogenic (P/LP) variants in genes that are not included in the ACMG SFs but have a clear medically actionable value (non-ACMG SF), suggesting expansion of the ACMG SF list.

# Results

## Cohort description

The study cohort consists of 863 Pakistani individuals (Table 1): 700 (81.1%) index participants and 163 (19.9%) family members, with 532 (61.7%) males. Most participants, 741 (85.9%), were clinically affected and had various symptoms; their median age was 3 yr; the median age of 122 (14.1%) healthy/asymptomatic participants was 29.5 yr. 559 (79.9%) participants had consanguineous parents, with most of them self-declaring ethnicity as Punjabi.

## SFs

### SFs of the ACMG v3.1 gene list (ACMG SFs)

24 ACMG SFs were detected in 23 of 863 (2.7%) participants. One participant had two ACMG SFs (in MYBPC3 and TTN genes). Two members of one family had the same variant in TMEM43. Two unrelated participants have the same variant in KCNQ1. There were 23 monoallelic (heterozygous) and 1 (4.2%) biallelic (homozygous) mutations. In total, there were 22 unique variants in 14 different genes associated with 15 genetic diseases. Only one ACMG SF was found in an asymptomatic participant (in RYR1), giving a frequency of 0.8%, whereas the frequency of ACMG SFs in symptomatic participants was 3.0% (OR 3.6 [95% CI: 0.5–27.1], P = 0.21) Table S1). 18/24 (75.0%) ACMG SFs were related to cardiovascular diseases (CVD), among them: TTN dilated cardiomyopathy (DCM)—4; ACTC1 DCM (1); MYBPC3 hypertrophic cardiomyopathy (HCM)—2, TNNT2 DCM/HCM—1; KCNQ1 long QT syndrome (LQTS)—3; LDLR familial hypercholesterolemia (FH)—2; PCSK9 FH—1; APOB FH—1; TMEM43 arrhythmogenic right ventricular dysplasia—2; FBN1 Marfan syndrome (MS)—1. Gene-related cancer predisposition syndromes accounted for 16.7% (4/24) of ACMG SFs: MSH6 Lynch syndrome—1, BRCA1 hereditary breast and ovarian cancer (HBOC)—1, and PALB2 hereditary breast cancer (HBC)—2. Pathogenic variants in RYR1 related to malignant hyperthermia (MH) were detected in two (8.3%) participants (Fig 1).

We detected one participant (0.12%) with a heterozygous likely pathogenic (LP) variant in the MUTHYH gene, meaning a carrier status for the autosomal recessive (AR) MUTYH-related familial adenomatous polyposis.

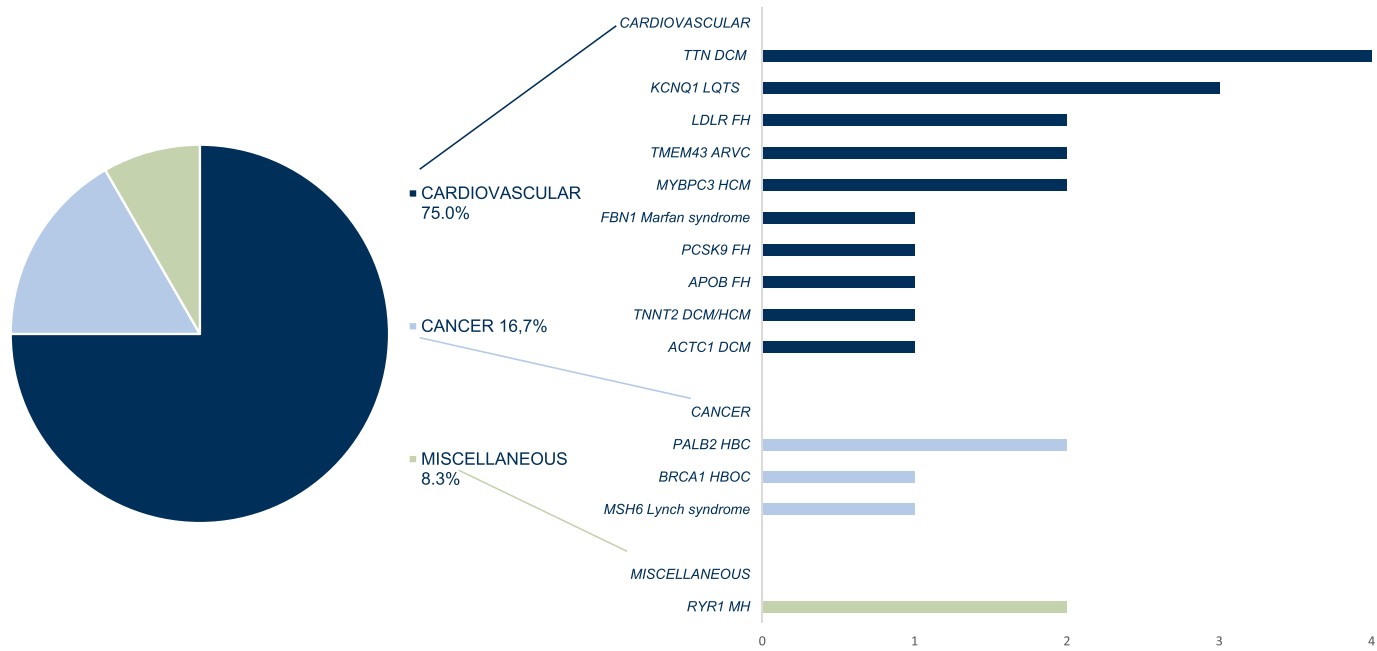

**Figure 1. The ACMG v3.1 secondary findings.**
Disease categories and related gene-disease pairs. The ACMG, the American College of Medical Genetics and Genomics; DCM, dilated cardiomyopathy; LQTS, long QT syndrome; FH, familial hypercholesterolemia; ARVC, arrhythmogenic right ventricular cardiomyopathy; HBC, hereditary breast cancer; HBOC, hereditary breast and ovarian cancer; MH, malignant hyperthermia.

### SFs that are not included in the ACMG SF (v3.1) list (non-ACMG SFs)

In addition to ACMG SFs, we analysed other P/LP variants in genes that were not related to the participants' clinical conditions but had clear medical actionability—non-ACMG SFs (see the Materials and Methods section).

16 non-ACMG SFs were detected in 16 (1.9%) of 863 participants. There were 8/16 (50%) monoallelic (heterozygous and hemizygous) and 8/16 (50%) biallelic (homozygous and compound heterozygous) mutations. Totally, there were 17 unique variants in 14 unique genes related to 15 genetic diseases (Table S1). Among non-ACMG SF 4/16 (25%) were related to eye disease, followed by metabolic, 2/16 (12.5%), and urinary system diseases, 2/16 (12.5%); CVD accounted for 6.25% (1/16) (Fig 2). We detect non-ACMG SFs in 15/741 (2.0%) symptomatic and 1/122 (0.8%) asymptomatic participants, with an OR of 0.40 (95% CI: 0.05–3.09, P = 0.38) (Table S1).

Eight participants (0.93%) had a carrier status and heterozygous P/LP variants for AR diseases: *CFTR* cystic fibrosis (5), *ITGB4* junctional epidermolysis bullosa type 5A (1), *MEFV* AR familial Mediterranean fever (1), and *SLC7A9* AR cystinuria (1) (Table S1).

### PFs

According to the phenotype and family history of the participants, some gene-disease pairs included in both the ACMG SF and non-ACMG SF lists were reported as PFs (ACMG PFs and non-ACMG PFs, respectively).

### PFs of the ACMG v3.1 (2022) SF list (ACMG PF)

The ACMG PFs were detected in 41/741 (5.5%) symptomatic participants. For comparison to the ACMG SFs, we excluded those with variants of uncertain significance (VUS). The ACMG PFs that included only P/LP variants were detected in 35/741 (4.7%) symptomatic participants. The same homozygous variants were found among two family members in three families: (1) and (2) in the *BTD* gene; and (3) in the *ATP7B* gene. There were 4 (11.4%) monoallelic (heterozygous) and 31 (88.6%) biallelic (homozygous and compound heterozygous) mutations among the ACMG PFs. Thus, in contrast to the ACMG SFs, biallelic variants predominated among the ACMG PFs. In total, there were 23 unique P/LP variants in eight genes related to eight genetic diseases (Table S2).

Most gene-disease pairs belong to either only the ACMG SF or only the ACMG PF (Fig 3). The most frequent in the ACMG SFs were gene/disease and frequency *TTN* DCM (four times); *KCNQ1* LQTS (three times); MYBPC3 HCM (two times); *TMEM43* arrhythmogenic right ventricular cardiomyopathy (two times); *PALB2* HBC (two times); and *RYR1* MH (two times). The most frequent in the ACMG PFs were *BTD* biotinidase deficiency (14), *APT7B* Wilson disease (11), and *GAA* Pompe disease (4). Two gene-disease pairs occurred in both groups, the ACMG SF and PF: *FBN1* MS (one time as SF, one time as PF); and *LDLR* FH type 1 (two times as SF, two times as PF). No identical variants have been found in both groups (Tables S1 and S2).

### PFs of non-ACMG SF list (non-ACMG PF)

The non-ACMG PFs were detected in 11/741 (1.5%) symptomatic participants. The non-ACMG PFs that included only P/LP variants were detected in 10/741 (1.3%) individuals. Compound heterozygous variants LP/VUS in *ABCA4* gene were found in one participant. We included this participant in further analysis, but *ABCA4* VUS was excluded from variant analysis. We found only biallelic (homozygous and compound heterozygous) variants among the non-ACMG

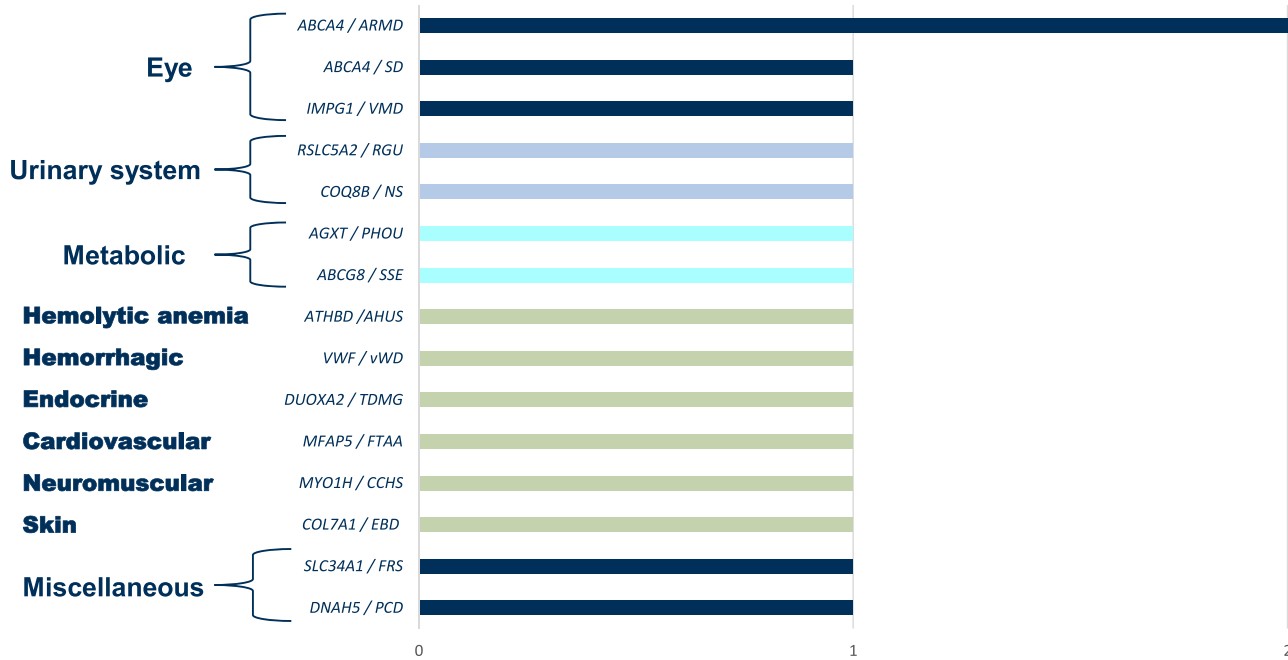

**Figure 2.   Non-ACMG secondary finding.**
Disease categories and related gene-disease pairs. The ACMG, the American College of Medical Genetics and Genomics; ARMD, age-related macular degeneration type 2; SD, Stargardt disease type 1; PCD, primary ciliary dyskinesia type 3; FRS, Fanconi renotubular syndrome type 2; VMD, vitelliform macular dystrophy type 4; CCHS, congenital central hypoventilation syndrome type 2; SSE, sitosterolemia type 1; PHOU, primary hyperoxaluria type 1; NS, nephrotic syndrome; type 9; RGU, renal glucosuria; FTAA, familial thoracic aortic aneurysm type 9; TDMG, thyroid dyshormonogenesis type 5; vWD, von Willebrand disease type 1; AHUS, atypical hemolytic uremic syndrome type 6; EBD, epidermolysis bullosa dystrophica.

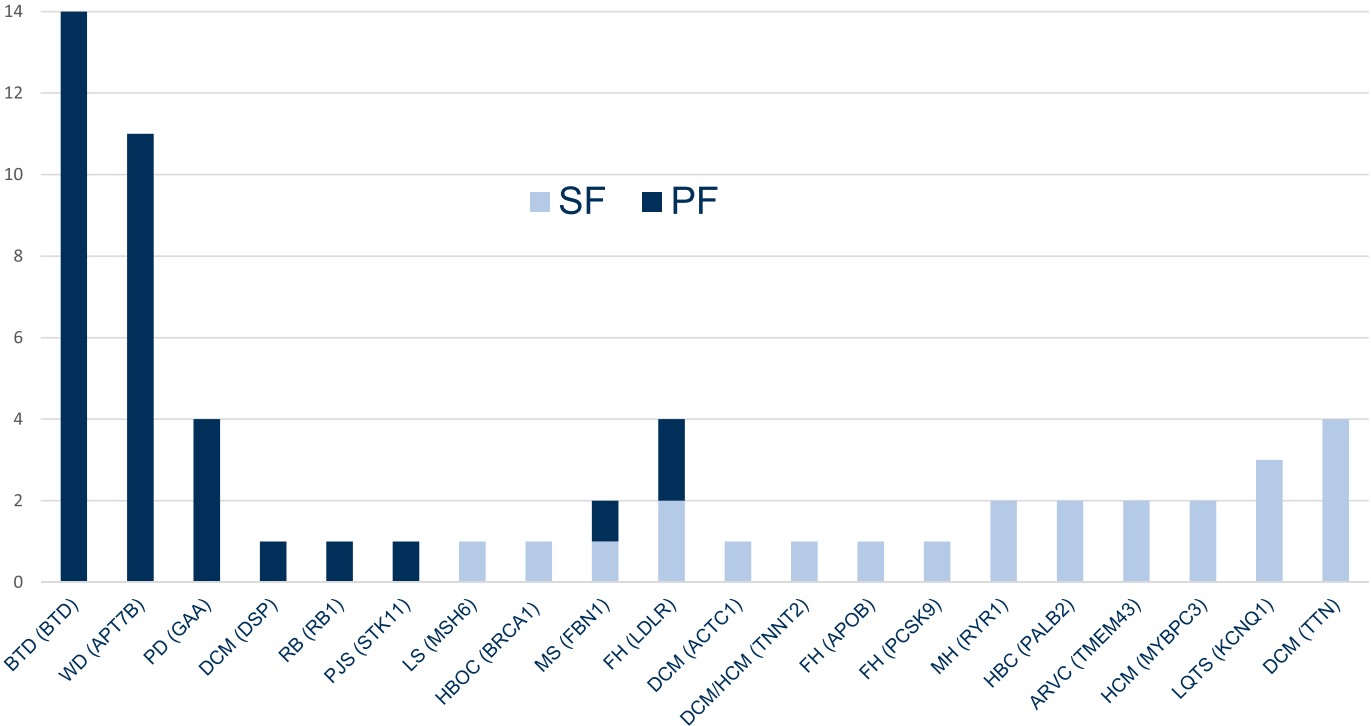

**Figure 3.   The ACMG secondary findings and primary findings.**
Frequency of (gene)-disease pairs. The ACMG, the American College of Medical Genetics and Genomics; BTD, biotinidase deficiency; WD, Wilson disease; PD, Pompe disease; DCM, dilated cardiomyopathy; RB, retinoblastoma; PJS, Peutz-Jeghers syndrome; LS, Lynch syndrome; HBOC, hereditary breast and ovarian cancer; MS, Marfan syndrome; FH, familial hypercholesterolemia; HCM, hypertrophic cardiomyopathy; MH, malignant hyperthermia; HBC, hereditary breast cancer; ARVC, arrhythmogenic right ventricular cardiomyopathy; LQTS, long QT syndrome.

PFs (100%). In total, there were 11 unique P/LP variants in three genes related to three genetic diseases: *COL7A1* AR Epidermolysis bullosa dystrophica, *ABCA4* Stargardt disease type 1, and CFTR cystic fibrosis (Table S2). Only one gene-disease pair occurred in both groups, the non-ACMG SF and PF: *ABCA4* Stargardt disease type 1 (one time as SF, two times as PF). No identical variants were found in both groups (Tables S1 and S2).

## Discussion

In our study, we aimed to define the frequency of secondary genomic findings based on the ACMG v3.0 (2021) gene list and the distribution of the associated disease type in the Pakistani population. Furthermore, based on criteria described in the methods section, we propose the inclusion of additional genes to the SF list. Considering the ACMG gene-disease pairs, our results demonstrated a prevalence of 2.7% SFs in the Pakistani population.

The rates of ACMG SFs in previous studies ranged from 0.6% (Jain et al, 2018) to 6.6% (Jang et al, 2015). In the study of Aloraini et al (Aloraini et al, 2022), the overall rate of SFs in the Saudi population was higher than 8%. However, the rate of SF findings in 48 individuals (family members with the same SF variants were excluded) was calculated for the number of families (574) participated in the study. It is perceivable that the SF rate will significantly be reduced if all participants (1,254) would have been included in the calculation.

The number of participants included in the SF studies ranged from less than 200, 161 (Kuo et al, 2020), 196 (Jang et al, 2015), or less than 300, 280 (Jalkh et al, 2020), to 1,000, 6,240 (Elfatih et al, 2021), and 21,915 (Gordon, 2020). The studies with very large sample size can present some hurdles, such as ethical or financial, however, studies with too small sample sizes may prevent the findings from being extrapolated to population level (Faber & Fonseca, 2014). Among the published studies, the highest frequencies of SFs, 6.6% and 6.1%, were shown in the studies with relatively low numbers of participants, 196 and 280, respectively (Jang et al, 2015; Jalkh et al, 2020).

Most of the SF studies represented local populations including Saudi Arabia (Aloraini et al, 2022), China (Chen et al, 2018), Qatar (Jain et al, 2018; Elfatih et al, 2021), Lebanon (Jalkh et al, 2020), Korea (Jang et al, 2015), Taiwan (Kuo et al, 2020), Netherlands (Haer-Wigman et al, 2019), Singapore (Jamuar et al, 2016), and Thailand (Chetruengchai & Shotelersuk, 2021). There were more representative studies, with self-reported race/ethnic groups including Hispanic or Latinx, black or African American, Asian, American Indian, Alaska Native, or Pacific Islander, and white (Gordon, 2020); East Asian ancestry (China, Vietnam) (Tang et al, 2018), European, and African Americans (Natarajan et al, 2016), and with 1,000 Genomes Project, which includes 14 different populations in four major ancestry groups (Europe, East Asia, Africa, and the Americas) (Olfson et al, 2015). So far, to the best of our knowledge, no studies representing the Pakistani population and its ethnic groups have been conducted.

Depending on the time of publication, the studies used the v2.0 update of ACMG gene list, 2,016 (59 genes) (Chen et al, 2018; Jain et al, 2018; Tang et al, 2018; Haer-Wigman et al, 2019; Gordon, 2020; Jalkh et al, 2020; Kuo et al, 2020; Elfatih et al, 2021; Aloraini et al, 2022)

or ACMG gene list v1.0, 2,013 (56 genes) (Jang et al, 2015; Olfson et al, 2015; Jamuar et al, 2016; Natarajan et al, 2016; Hart et al, 2019). To our knowledge, only one study was conducted to determine the frequency of P/LP variants in the 73 genes of ACMG v3.0 SF, in which a SF frequency of 5.5% was reported (Chetruengchai & Shotelersuk, 2022). Studies using ACMG v3.1 list have not yet been conducted. Considering the small difference in the number of genes between ACMG lists v1.0 and v2.0 (56 versus 59), a significant increase in SFs was not expected. However, because 14 additional genes were added to the ACMG v3.0 and 5 additional genes were added to the ACMG 3.1 SF gene list, we addressed whether this upgrade would affect the frequency and change the disease structure of SFs in future studies. In our study, the participants with a variant in *TTN* gene, exclusively present in v3.0 and v3.1 SFs, resulted in an increase of 12.5%: 2.7% v3.1 versus 2.4% v2.0 ACMG SF lists.

Disease-causing copy number variants (CNV) account for 5–9% of hereditary diseases (Lindy et al, 2018; Truty et al, 2019; Brandt et al, 2020). Although CNV can be identified by WES, the specificity and sensitivity limit their detection to larger alteration, affecting at least three consecutive coding-targeted exons. Most SF studies used exclusively WES (Jang et al, 2015; Natarajan et al, 2016; Chen et al, 2018; Haer-Wigman et al, 2019; Kuo et al, 2020; Aloraini et al, 2022) or both WES and WGS (Olfson et al, 2015; Jamuar et al, 2016; Jain et al, 2018; Hart et al, 2019). Only few SF studies used exclusively WGS for testing all individuals (Tang et al, 2018; Elfatih et al, 2021). Studies that used WGS can demonstrate a higher rate of SFs compared with those that used WES. For example, the substantial difference in the results of the two studies from the same country, Qatar, is probably in part because of the difference in the sequencing methods used. A predominantly WES (in 917 of 1,005 participants, 97%) was used in the first study published in 2018 with the SF rate of 0.6% (Jain et al, 2018). In the second study, 2021, WGS alone performed in all 6,045 participants showed 2.3% of v2.0 ACMG SF (Elfatih et al, 2021). In fact, CNV SF variants would have been missed with WES.

Only one study reported the proportion of consanguinity rate (60%) among its participants (Aloraini et al, 2022). In our study, 79.9% of participants were known to have consanguineous parents.

In most studies, SFs were found as monoallelic mutations. Aloraini et al, in their study, with 60% of consanguinity rate reported 1/49 (2.0%) LP variant biallelic (homozygous) in gene-disease pairs with AR mode of inheritance (Aloraini et al, 2022). In our study, 1/24 (4.2%) homozygous pathogenic variant (in *LDLR* gene) was found among ACMG SF and 8/16 (50.0%) biallelic (homozygous and compound heterozygous) mutations were found among non-ACMG SF.

The studies on SF included symptomatic individuals for whom sequencing was performed for diagnostic purposes (Natarajan et al, 2016; Tang et al, 2018; Jalkh et al, 2020; Kuo et al, 2020), or both symptomatic and asymptomatic individuals (Jang et al, 2015; Jamuar et al, 2016; Chen et al, 2018; Gordon, 2020; Elfatih et al, 2021; Aloraini et al, 2022). Two studies included exclusively healthy/asymptomatic participants (Jain et al, 2018; Haer-Wigman et al, 2019). The studies with only symptomatic individuals reported the rate of SF ranged from 1.0% (Natarajan et al, 2016) to 6.1% (Jalkh et al, 2020). The studies with only asymptomatic individuals reported the rate of SF ranged from 0.6% to 2.7% (Jain et al, 2018; Haer-Wigman et al, 2019).

One study compared SF frequency among symptomatic and asymptomatic individuals, showing no difference: 7% for the healthy subjects (7/100) and 6% for the patients with a disease (6/96) (Jang et al, 2015). In our study, the rates of SF among symptomatic were higher, though not significantly, for both ACMG SF (3.0% versus 0.8%) and non-ACMG SF (2.0% versus 0.8%).

In previous SF studies, PFs were excluded from further analysis. In our study, we further analysed PFs and compare them to SFs. In contrast to the SFs, biallelic variants predominated among the PFs: 88.6% in ACMG PFs and 100% in non-ACMG PFs. Most gene-disease pairs were either only SF or only PF in both groups. Only two gene-disease pairs occurred in both groups, ACMG SF and PF: *FBN1* MS (one time as SF, one time as PF); and *LDLR* FH type 1 (two times as SF, two times as PF). No identical variants in these genes have been found in both groups (Fig 3). Only one gene-disease pair occurred in both the non-ACMG SF and PF groups: *ABCA4* Stargardt disease type 1 (one time as SF, two times as PF). No identical variants were found in both groups (Tables S1 and S2). We can conclude that diseases presenting as PF only (e.g., Wilson disease, biotinidase deficiency) are more clinically severe with earlier onset of manifestations. Whereas, other diseases detected as SF only (e.g., *TTN*- and *ACTC1*-related DCM) are less clinically severe and have later manifestation onset. The absence of the identical variants in *FBN1*, *LDLR*, and *ABCA4* genes associated simultaneously with both SF and PF may indicate that the clinical severity and time of onset of these diseases may depend on characteristics of the variants.

In previous ACMG SF studies, CVD and cancer predisposition conditions (CPC) were the leading disease types. The proportion of CVD ranged from 11.7% (Jalkh et al, 2020) to 54.5% (Haer-Wigman et al, 2019), and the proportion of CPC ranged from 16.7% (Chen et al, 2018) to 46.2% (Jang et al, 2015). High percentage of CVD (75.0%), in our study, can be partly explained by different ACMG versions used. Using v3.0 we included four *TTN* variants. *TTN* cardiomyopathy is absent in previous versions of ACMG SF lists. This substantially increased the percentage of CVD and decreased the percentage of other conditions.

In the SF studies among CVD Brugada syndrome, LQTS and HCM were reported most frequently. BRCA1/2 HBOC and Lynch syndrome were most often reported cancer syndromes. Other frequently reported SFs include FH, MH, MS, and Loeys-Dietz syndrome. (Jang et al, 2015; Olfson et al, 2015; Jamuar et al, 2016; Natarajan et al, 2016; Chen et al, 2018; Jain et al, 2018; Tang et al, 2018; Haer-Wigman et al, 2019; Hart et al, 2019; Gordon, 2020; Jalkh et al, 2020; Kuo et al, 2020; Elfatih et al, 2021; Aloraini et al, 2022). In general, the structure of diseases in the SF mainly reflects the prevalence of diseases in the population. The diseases with high prevalence were detected with higher frequency: Brugada syndrome—0.5% per 1,000 (Vutthikraivit et al, 2018), HCM—2–6% per 1,000 (Batzner et al, 2019), HBOC—2.9–26.5% (Armstrong et al, 2019), and Lynch syndrome—0.44% (Haraldsdottir et al, 2017).

In our study, *TTN* DCM type 1G, *KCNQ1* LQTS type 1, *MYBPC3* HCM type 4, and *TMEM43* arrhythmogenic right ventricular dysplasia type 5 were the most frequent CVD, and *MSH6* Lynch syndrome and *PALB2* HBC were the most frequent CPC among ACMG SFs (Fig 1). *ABCA4* age-related macular degeneration type 2 was most frequent among non-ACMG SFs (Fig 2). We can assert that these diseases are frequent among the population of Pakistan. The overall prevalence of CAD in Pakistan ranges from 22% to 32% (Jafar et al, 2005). The

prevalence of various cancers in Pakistan is high: the prevalence of breast cancer in females ranges from 20% to 50%, and the prevalence of colorectal cancer in both males and females ranges from 4% to 6% (Idrees et al, 2018).

The studies reported individuals to be carriers of a disease allele in genetic disorders with AR mode of inheritance from the ACMG SF list v2.0 with rates ranged from 0.40% (Jain et al, 2018; Elfatih et al, 2021) to 3.1% (Kuo et al, 2020). Despite the increase of number of AR diseases in ACMG v3.1 SF list, a smaller proportion of carriers (0.12%) were identified in out cohort. This could be because of low proportion of healthy individuals in our cohort compared with the studies mentioned above.

## Limitations of our study

The study cohort consists predominantly of symptomatic young children, and we refrain from directly extrapolating the results to the population of Pakistan as a whole. Furthermore, the generation of the non-ACMG SF gene list is mainly based on the opinion of experts, which may have limited the number of genes.

## Conclusions

WGS is a reliable and easy test format to identify the ACMG 3.1 and the non-ACMG SF. The frequency of ACMG SF (2.7%) is within the range reported in most relevant studies. Higher proportion of CVD (75.0%) among ACMG SFs in our study can partly be explained by the additional inclusion of CVD (e.g., *TTN* DCM) in the ACMG v3.0 and v3.1 SF list. In addition, we reported a 1.9% rate of Pakistan-specific non-ACMG SF and an unexpectedly high proportion of biallelic variants among both the ACMG SF (4.2%) and the non-ACMG SF (50%). These results are relevant to the epidemiology of Pakistan as a country with a high rate of consanguineous marriages. Our findings may help guide the development of standards of practice in genomic medicine. As such, in countries with high levels of consanguinity, the ACMG criteria for SF can be expanded, and our list of non-ACMG SF is one example.

# Materials and Methods

## Participants

All participants were tested as part of clinical evaluation in Pakistani hospitals in Islamabad, Karachi, Lahore, Multan, and Peshawar for different reasons. Subjects were either index cases or healthy family members. The participants or guardians of the children participants signed the provided written informed consent forms. The study was approved by the Ethics Committee of Rostock University, A2022-0072, 25.04 2022.

## WGS and data analysis

DNA samples were prepared using the TruSeq DNA Nano Library Prep Kit from Illumina. The libraries were pooled and sequenced with the 150-bp paired-end protocol on an Illumina platform to

yield an average coverage depth of 30× for the nuclear genome. Raw read alignment to reference genome GRCH38 and variant calling, including single nucleotide substitutions (SNVs), small insertions/deletions (Indels), and structural variants (SVs) with default parameters, were performed using DRAGEN (version 3.10.4, Illumina). SNV and indel annotation were performed by Varvis (Limbus Medical Technologies GmbH; https://www.limbus-medtec.com/). Structural variants were annotated with ANNOTSV3.1 and the in-house structural variant database to obtain occurrence frequencies. Genetic variants are described after the Human Genome Variation Society recommendations (https://varnomen.hgvs.org/).

**Variant evaluation and interpretation**

Only good-quality variants with a minimum of nine reads and an alternate allele frequency of at least 0.3% were considered. Candidate variants were evaluated with respect to their pathogenicity and causality and categorized following ACMG guidelines (Richards et al, 2015) using the five-tier classes: pathogenic, LP, variants of uncertain significance (VUS), likely benign, and benign. Variants were assessed in a routine diagnostic setting. An initial analysis was restricted to genes that have a clear association with the participant's phenotype using Human Phenotype Ontology nomenclature (https://hpo.jax.org/app/). The variants found in these genes were "the primary findings." During this step, common standards were followed. Briefly, the following aspects were considered: the minor frequency of the allele in control databases (gnomAD) and internal database; the in silico pathogenicity prediction and potential impact on respective proteins; the known mechanism of the variant type-disease (missense, truncating, etc.); segregation in the family; and available external evidence: OMIM (https://www.omim.org/), ClinVar information (https://www.ncbi.nlm.nih.gov/clinvar/), Mastermind (https://mastermind.genomenon.com/), and genotype-phenotype correlation.

SFs that do not correlate with the provided phenotype(s) were reported according to ACMG recommendations version 3.1 (2022) for reporting of SFs in clinical exome and genome sequencing (Miller et al, 2022).

In addition, we reported SFs from the list of non-ACMG SFs. For this approach, ACMG and non-ACMG genes were interrogated, and participant's phenotype was not considered. Only pathogenic and LP variants with appropriate zygosity according to the known mode of inheritance for the corresponding gene (i.e., heterozygous for AD and homozygous or compound heterozygous for AR) were considered for reporting. Finally, the data were evaluated by screening the carrier status for both ACMG and non-ACMG-SF genes, including only heterozygous genetic variants, classified as pathogenic or LP for AR diseases.

Arguments for reporting SFs or not doing so relate to the principles of autonomy, non-maleficence, and beneficence. Patient's informed consent to report SFs was a mandatory element of the ethics of this study. Patient's informed consent was obtained from all participants.

***SFs that were not included in the ACMG SF v3.1 list (non-ACMG SFs)***
The list containing the non-ACMG SF gene-disease pairs (non-ACMG SF), was generated based on our internal medical expertise in the field, and validated by several well-known and broadly used external sources (ClinGen, https://clinicalgenome.org/; eMERGE, https://emerge-network.org/). During the selection process, we have used the following criteria: disease severity; likelihood of disease; effectiveness of intervention; and nature of intervention. The score was calculated as follows: disease severity: three—sudden death, two—possible death or major morbidity, one—modest morbidity, 0—minimal or no morbidity; likelihood of disease: 3—>40% chance, 2—5–39% chance, 1—1–4% chance, 0—<1% chance or unknown; effectiveness of intervention: 3—highly effective, 2—moderately, 1—minimally, 0—ineffective/no intervention; nature of intervention: 3—low risk/medically acceptable/low intensity intervention, 2—moderately acceptable/risk/intensive intervention, 1—greater risk/less acceptable/substantial intervention, 0—high risk/poor acceptable/intensive/or no intervention. Level of evidence was also added to likelihood of disease and effectiveness of intervention: A—substantial evidence; B—moderate; C—minimal; D—poor; N—non-systematically or expert contributed evidence. From our list which initially contained 154 gene-disease pairs with score of 10 or more, we have finally selected 39 gene-disease pairs. The entire list of non-ACMG SF with actionability score is shown in Table S3.

# Data Availability

The variant from this study have been submitted to the NCBI ClinVar database (http://www.clinvar.com/) under accession numbers SCV002564547–SCV002564628.

# Supplementary Information

# Acknowledgements

We thank the contributions of the administration and staff of the Children Hospital (Lahore), Pakistan Institute of Medical Sciences (Islamabad), Children Hospital and ICH (Multan), Town Women and Children Hospital (Peshawar), Liaquat National Hospital (Karachi), and Arcensus laboratory and bioinformatics teams.

**Author Contributions**

A Skrahin: conceptualization, data curation, formal analysis, validation, investigation, visualization, methodology, and writing—original draft, review, and editing.
HA Cheema: data curation, investigation, and writing—review and editing.
M Hussain: investigation and writing—review and editing.
NN Rana: investigation and writing—review and editing.
KU Rehman: investigation and writing—review and editing.
R Kumar: investigation and writing—review and editing.

G Oprea: resources, data curation, formal analysis, validation, investigation, visualization, methodology, and writing—original draft, review, and editing.

N Ameziane: resources, data curation, formal analysis, validation, investigation, visualization, methodology, and writing—original draft, review, and editing.

A Rolfs: conceptualization, resources, data curation, formal analysis, supervision, investigation, methodology, and writing—original draft, review, and editing.

V Skrahina: conceptualization, resources, data curation, formal analysis, supervision, investigation, methodology, project administration, and writing—original draft, review, and editing.

## Conflict of Interest Statement

A Skrahin, A Rolfs, G Oprea, N Ameziane, and V Skrahina declared employment in genomic testing company; HA Cheema, M Hussain, NN Rana, KU Rehman, and R Kumar did not declare any competing interest related to this study.

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
