## [Reviewer comments · Life Science Alliance]

Life Science Alliance

Secondary findings in a large Pakistani cohort tested with whole genome sequencing

Aliaksandr Skrahin, Huma Cheema, Maqbool Hussain, Nuzhat Rana, Khalil Rehman, Raman Kumar, Gabriela Oprea, Najim Ameziane, Arndt Rolf, and Volha Skrahina

DOI: <https://doi.org/10.26508/lsa.202201673>

Corresponding author(s): Aliaksandr Skrahin, Arcensus GmbH

Review Timeline:

Submission Date:	2022-08-16
Editorial Decision:	2022-10-05
Revision Received:	2022-11-21
Editorial Decision:	2022-12-08
Revision Received:	2022-12-21
Accepted:	2022-12-21

Scientific Editor: Novella Guidi

Transaction Report:

October 5, 2022

Re: Life Science Alliance manuscript #LSA-2022-01673-T

Aliaksandr Skrahin
Arcensus GmbH Goethestr. 20 18055 Rostock, Germany
Goethestr. 20
Rostock 18055
Germany

Dear Dr. Skrahin,

Thank you for submitting your manuscript entitled "Secondary findings in a large Pakistani cohort tested with whole genome sequencing" to Life Science Alliance. The manuscript was assessed by expert reviewers, whose comments are appended to this letter. We invite you to submit a revised manuscript addressing the Reviewer comments.

Thank you for this interesting contribution to Life Science Alliance. We are looking forward to receiving your revised manuscript.

Sincerely,

B. MANUSCRIPT ORGANIZATION AND FORMATTING:

Reviewer #1 (Comments to the Authors (Required)):

Comments to the Author

In this article, the authors presented a compelling report of clinically actionable secondary findings on genetic variants identified by whole genome sequencing from a unique population. This Pakistani cohort includes index cases and healthy family members. The paper is an important contribution to research to improve publicly available genomic data diversity. However, the findings are similar to previously published surveys of the ACMG genes in exome and genome data. Overall, the strengths of this article include the unique population, the size and scope of the whole genome sequencing results, the identification of variants that are "clinically actionable" based on current American College of Medical Genetics guidelines, and the inclusion of non-ACMG SF list that could be specific to the characteristic of the studied population. However, there are several limitations/considerations which may further strengthen this article.

- The terms "Primary findings" and "Secondary Findings" should be clarified and adequately defined.
- The criteria used to analyze the potentially clinically significant non-ACMG variants should be better explained. The cohort of about 1000 Pakistani participants was used!! Is it a separate cohort of disease or healthy participants? What type of analysis was conducted on this cohort? What it's relevant to the studied cohort?
- An important point that needs to be brought out is the frequency of actionable variants for each diagnosis (and perhaps in aggregate). For example, it would be helpful to present the frequency of individuals who are genotype positive for an incidentally identified variant deemed LP/P and a corresponding frequency of those who also demonstrated evidence of that disease implicated by the variant.
- In Figure 1 and the manuscript, removing the variants associated with Familial hypercholesterolemia and Marfan syndrome would be better from the cardiovascular diseases list.
- Is it possible to include the GnomAD overall allele frequency for these variants to allow a reader to judge which variants appear to be unique or increased in frequency in the Pakistani population?

Reviewer #2 (Comments to the Authors (Required)):

In study by Skrahin et al 863 individuals from Pakistan were sequenced to study secondary findings. Pakistan is a developing country with very high proportion of consanguine marriages. Therefore genotyping that many individuals represents a very interesting and important effort and could help in treatment of many people. However, I find that the extensive dataset produced in this study wasn't used efficiently.

For comparison, one could take an earlier similar study with shared co-authors (Cheema et al 2020 <https://www.nature.com/articles/s41525-020-00150-z>, for some reason not cited in the current study), where clinical utility was clearly indicated.

In the current study results are presented in a very descriptive manner and basically summarize the numbers of gene-disease pairs. Many questions, which could be addressed with the sequenced data stay unanswered. Like for example, how the inbreeding coefficient would affect the frequencies of primary and secondary findings? Or how the gene regulatory elements are affected? One could imagine lots of other interesting questions, which could be immediately addressed in this kind of study without applying too much effort.

Since the results presented in the manuscript are purely descriptive, on the technical side there are just minor comments concerning presentation of data and manuscript language:

- 1) Instead of percentages of different categories of secondary findings, it is better to just indicate the actual numbers (like out of 24, 18 related to cardiovascular diseases and so on). Percentages are misleading when the numbers are so low.

2) For clinically affected patients, age SD exceeds the mean age, indicating a very abnormal distribution. Median should be provided in that case.

3) Multiple grammar errors in text should be taken care of. For example:

"the of use of different sequencing methods" (page 3).

"we propose the inclusion of addition genes the SF list" (page 10).

Reviewer #1:

In this article, the authors presented a compelling report of clinically actionable secondary findings on genetic variants identified by whole genome sequencing from a unique population. This Pakistani cohort includes index cases and healthy family members. The paper is an important contribution to research to improve publicly available genomic data diversity. However, the findings are similar to previously published surveys of the ACMG genes in exome and genome data. Overall, the strengths of this article include the unique population, the size and scope of the whole genome sequencing results, the identification of variants that are "clinically actionable" based on current American College of Medical Genetics guidelines, and the inclusion of non-ACMG SF list that could be specific to the characteristic of the studied population.

Answer: We are grateful to the reviewer for appreciating the importance of our study, emphasizing its strengths.

Reviewer #1:

However, there are several limitations/considerations which may further strengthen this article. The terms "Primary findings" and "Secondary Findings" should be clarified and adequately defined.

Answer: We have added extended definitions for "primary findings" and "secondary findings" in the Introduction (page 3, lines 2-14) and Methods (page 14, lines 13-16 and lines 23-26; page 15, lines 1-3) sections.

Reviewer #1:

The criteria used to analyze the potentially clinically significant non-ACMG variants should be better explained. The cohort of about 1000 Pakistani participants was used!! Is it a separate cohort of disease or healthy participants? What type of analysis was conducted on this cohort? What it's relevant to the studied cohort?

Answer: We have analysed only one Pakistani cohort. To avoid any confusion the reader might encounter, we have improved our language and provide more information under the Methods section (page 15, lines 12-26; page 16 lines 1-2)

Reviewer #1:

An important point that needs to be brought out is the frequency of actionable variants for each diagnosis (and perhaps in aggregate). For example, it would be helpful to present the frequency of individuals who are genotype positive for an incidentally identified variant deemed LP/P and a corresponding frequency of those who also demonstrated evidence of that disease implicated by the variant.

Answer: We have added the observed internal allele frequency to Supplement Tables S1 and S2.

Reviewer #1:

In Figure 1 and the manuscript, removing the variants associated with Familial hypercholesterolemia and Marfan syndrome would be better from the cardiovascular diseases list.

Answer: We agree that Marfan syndrome is a systemic connective tissue disorder and familial hypercholesterolemia is a metabolic disorder. However, we have categorized these as cardiovascular based on most recent guidelines to report secondary findings (Miller et al. 2021; Miller et al. 2022).

Reviewer #1:

Is it possible to include the GnomAD overall allele frequency for these variants to allow a reader to judge which variants appear to be unique or increased in frequency in the Pakistani population?

Answer: We have included the GnomAD overall allele frequency for the variants to allow readers to judge which variants appear to be unique or increased in frequency in the Pakistani population (Supplement Tables S1 and S2).

Reviewer #2:

In study by Skrahin et al 863 individuals from Pakistan were sequenced to study secondary findings. Pakistan is a developing country with very high proportion of consanguine marriages. Therefore, genotyping that many individuals represents a very interesting and important effort and could help in treatment of many people.

Answer: We thank the reviewer for this comment.

Reviewer #2:

However, I find that the extensive dataset produced in this study wasn't used efficiently. For comparison, one could take an earlier similar study with shared co-authors (Cheema et al 2020 <https://www.nature.com/articles/s41525-020-00150-z>, for some reason not cited in the current study), where clinical utility was clearly indicated.

Answer: We have included this publication in the Introduction section (page 4, lines 23-26; page 5, lines 1-5, as well as in the list of references (Cheema et al. 2020), and this has undoubtedly enriched our manuscript.

Reviewer #2:

In the current study results are presented in a very descriptive manner and basically summarize the numbers of gene-disease pairs. Many questions, which could be addressed with the sequenced data stay unanswered. Like for example, how the inbreeding coefficient would affect the frequencies of primary and secondary findings? Or how the gene regulatory elements are affected? One could imagine lots of other interesting questions, which could be immediately addressed in this kind of study without applying too much effort.

Answers: We agree with the reviewer's rational comment. Our manuscript leaves many questions unanswered. Since the descriptive character of our study was determined before the start of the study, the above questions unfortunately remained outside the scope of the study. We anticipate exploring these questions in our future research, and we are grateful to the reviewer for these valuable ideas.

Reviewer #2:

Since the results presented in the manuscript are purely descriptive, on the technical side there are just minor comments concerning presentation of data and manuscript language:

1) Instead of percentages of different categories of secondary findings, it is better to just indicate the actual numbers (like out of 24, 18 related to cardiovascular diseases and so on). Percentages are misleading when the numbers are so low.

Answer: Changed. We showed the results both in percentages and in actual numbers.

2) For clinically affected patients, age SD exceeds the mean age, indicating a very abnormal distribution. Median should be provided in that case.

Answer: Changed. We added median and range the Table 1.

3) Multiple grammar errors in text should be taken care of. For example:

"the of use of different sequencing methods" (page 3).

"we propose the inclusion of addition genes the SF list" (page 10).

Answer: Corrected. Other grammar errors also checked and corrected.

Cheema H, Bertoli-Avella AM, Skrahina V, Anjum MN, Waheed N, Saeed A, Beetz C, Perez-Lopez J, Rocha ME, Alawbathani S et al. 2020. Genomic testing in 1019 individuals from 349 Pakistani families results in high diagnostic yield and clinical utility. *NPJ Genom Med* **5**: 44.

Miller DT, Lee K, Abul-Husn NS, Amendola LM, Brothers K, Chung WK, Gollob MH, Gordon AS, Harrison SM, Hershberger RE et al. 2022. ACMG SF v3.1 list for reporting of secondary findings in clinical exome and genome sequencing: A policy statement of the American College of Medical Genetics and Genomics (ACMG). *Genet Med* **24**: 1407-1414.

Miller DT, Lee K, Chung WK, Gordon AS, Herman GE, Klein TE, Stewart DR, Amendola LM, Adelman K, Bale SJ et al. 2021. ACMG SF v3.0 list for reporting of secondary findings in clinical exome and genome sequencing: a policy statement of the American College of Medical Genetics and Genomics (ACMG). *Genet Med* **23**: 1381-1390.

December 8, 2022

RE: Life Science Alliance Manuscript #LSA-2022-01673-TR

Dr. Aliksandr Skrahin
Arcensus GmbH
Goethestr. 20
Rostock 18055
Germany

Dear Dr. Skrahin,

Thank you for submitting your revised manuscript entitled "Secondary findings in a large Pakistani cohort tested with whole genome sequencing". We would be happy to publish your paper in Life Science Alliance pending final revisions necessary to meet our formatting guidelines.

-please add a figure legend section to your manuscript, including your main figure legends and your table legends
-we could not find the data in ClinVar using the information provided. A ClinVar accession number (VCV, RCV, or SCV) would be useful.

A. FINAL FILES:

B. MANUSCRIPT ORGANIZATION AND FORMATTING:

****It is Life Science Alliance policy that if requested, original data images must be made available to the editors. Failure to provide**

original images upon request will result in unavoidable delays in publication. Please ensure that you have access to all original data images prior to final submission.**

The license to publish form must be signed before your manuscript can be sent to production. A link to the electronic license to publish form will be sent to the corresponding author only. Please take a moment to check your funder requirements.

Sincerely,

Reviewer #1 (Comments to the Authors (Required)):

The authors have addressed the critique

December 21, 2022

RE: Life Science Alliance Manuscript #LSA-2022-01673-TRR

Dr. Aliaksandr Skrahin
Arcensus GmbH Goethestr. 20 18055 Rostock, Germany
Goethestr. 20
Rostock 18055
Germany

Dear Dr. Skrahin,

Thank you for submitting your Research Article entitled "Secondary findings in a large Pakistani cohort tested with whole genome sequencing". It is a pleasure to let you know that your manuscript is now accepted for publication in Life Science Alliance. Congratulations on this interesting work.

DISTRIBUTION OF MATERIALS:

Again, congratulations on a very nice paper. I hope you found the review process to be constructive and are pleased with how the manuscript was handled editorially. We look forward to future exciting submissions from your lab.

Sincerely,
